# Orthologs at the Base of the Olfactores Clade

**DOI:** 10.3390/genes15060657

**Published:** 2024-05-22

**Authors:** Wilfred D. Stein

**Affiliations:** Silberman Institute of Life Sciences, Hebrew University, Jerusalem 91904, Israel; wdstein@mail.huji.ac.il

**Keywords:** evolution, orthologs, tunicates, olfactores, neural crest, type II cadherins, crystallins, connexins

## Abstract

Tunicate orthologs in the human genome comprise just 84 genes of the 19,872 protein-coding genes and 23 of the 16,528 non-coding genes, yet they stand at the base of the Olfactores clade, which radiated to generate thousands of tunicate and vertebrate species. What were the powerful drivers among these genes that enabled this process? Many of these orthologs are present in gene families. We discuss the biological role of each family and the orthologs’ quantitative contribution to the family. Most important was the evolution of a second type of cadherin. This, a Type II cadherin, had the property of detaching the cell containing that cadherin from cells that expressed the Type I class. The set of such Type II cadherins could now detach and move away from their Type I neighbours, a process which would eventually evolve into the formation of the neural crest, “the fourth germ layer”, providing a wide range of possibilities for further evolutionary invention. A second important contribution were key additions to the broad development of the muscle and nerve protein and visual perception toolkits. These developments in mobility and vision provided the basis for the development of the efficient predatory capabilities of the Vertebrata.

## 1. Introduction

In a 2010 paper, Domazet-Loso and Tautz [1] proposed a classification of the animal kingdom into 19 phylostrata, ranging from phylostratum 1, the first unicellular organisms, to phylostratum 19, the primates. In an alternative cladistic scheme—depicted in Figure 1–the phylostratum number is shown in parentheses after the name of the appropriate grouping:

The vertebrates and the tunicates share many of the genes that appeared over evolutionary time up to and including the emergence of the tunicates. Of these genes, those that were not present in earlier-appearing animals are the tunicate orthologs, the tunicate contributions to the vertebrate genome. By searching for orthologs, Domazet-Loso and Tautz [1] assigned many of the genes of the human genome to one or other of the 19 phylostratum levels. But finding orthologs is not an easy matter. Indeed, the Search for Orthologs Consortium comprises some dozen research groups that use various algorithms in their searches to handle the difficult problem of comparing tens of thousands of genes among thousands of living species. The results reported by these different research groups are by no means consistent. Liebeskind et al. [3] adopted a consensus approach where, for each gene, the modal value of the reports of the different research groups was chosen to represent that gene’s phylostratum level. Also reported was the step value, which is the median of the difference, for each gene, between the modal value and that reported by each ortholog research group. Litman and Stein [4] extended this work and published a list of all the protein-coding genes and many of the non-coding genes of the human genome, with each gene recorded with its phylostratum number.

The tunicates, with which this paper is concerned, fall in phylostratum 10, and the orthologs that they share with humans lie at the base of the Olfactores clade that consists of the tunicates and the vertebrates [2]. Phylostratum 10 in the Litman and Stein study [4] contained 84 protein-coding and 23 non-coding genes. These numbers are almost an order of magnitude smaller than the number of such genes found for any of the other phylostrata. We were intrigued by the fact that these small number of genes would lie at the base of the Olfactores clade, which branched out to the thousands of tunicate and vertebrate species. In what way did such a small number of genes begin that vast onward evolutionary process? But first we had to be sure that we had a valid set of tunicate orthologs and we proceeded to build a hand-curated set. Many of these orthologs are found to be present in gene families. In what follows, we will discuss these families in turn. The biological role of each family is described and the extent to which the tunicate orthologs contribute quantitatively to the family.

## 2. Methods

To prepare our list of tunicate-contributed orthologs, we began with the 84 protein-coding genes that Litman and Stein [4] had assigned to the tunicates, being phylostratum 10. Knowing that such an assignment was not absolute but only a consensus estimate, in that the various ortholog search groups did not agree with one another, we decided to hand-curate the list. Having only 84 genes to work with made the task of hand-curation feasible. Our searches for orthologs were performed using the protein BLAST (Basic Local Alignment Search Tool) program of the NCBI (National Center for Biotechnology Information of the National Library of Medicine): https://blast.ncbi.nlm.nih.gov/Blast.cgi (accessed on 6 September 2023) with the following search parameters: Max target sequences 1000, Expect threshold 200,000, Word size 2, Max matches in a query range 0, Matrix BLOSUM62, Gap Costs Existence: 11 Extension: 1, No compositional adjustments, No Low complexity regions filter.

To be recognised by us as an ortholog, a sequence found when searching the tunicates database had to be annotated with the same name as the probe sequence from the human genome. A sequence annotated as “-like” was rejected. In addition, the probe sequence had to be absent when the Branchiostomatidae database, or that of the Echinodermata or earlier-appearing clades were searched. In a few cases, noted explicitly in Appendix A, the tunicate sequence had been annotated as “name-a” instead of “name-1”, and similarly for “b”, “c”, and ”d”. In other cases, again explicitly listed in Appendix A, the tunicate sequence’s name was a more generic synonym but presented with a convincing Expect Value.

Sequence comparisons between two proteins and dot plots of the comparisons were made using the BLAST 2-sequences tool of the BLAST program, using the same search parameters as above. All one-on-one Expect values recorded in this paper were the results of a BLAST 2-sequences comparison between the two proteins.

Alignments between protein sequences were established and dot plots generated using the COBALT (Constraint-Based Alignment Tool) aligner at the NCBI https://www.ncbi.nlm.nih.gov/tools/cobalt/cobalt.cgi?CMD=Web (accessed on 6 September 2023). Phylograms were prepared using the phylogram tool of COBALT. Phylograms were rooted by explicitly determining this in the COBALT algorithm, at the gene with the lowest phylostratum number if there was such a gene, or left unrooted if there was no such case.

Properties of the orthologs listed in Appendix A were taken from the listings in GeneCards https://genealacart.genecards.org/Result (accessed on 7 August 2023). GeneCards also proved useful when the aliases of genes discussed in the literature had to be interpreted to provide HGNC symbols.

For finding the Unique Gene IDs of the tunicate-shared orthologs for Appendix A, we searched the ANISEED database at:

https://www.aniseed.fr/aniseed/gene/?choice=find_gene&module=aniseed&action=gene:index (accessed on 30 October 2023).

Gene Ontology data for the genes not found to be present in gene families was extracted from The Database for Annotation, Visualization and Integrated Discovery (DAVID) at https://david.ncifcrf.gov/summary.jsp (accessed on 31 August 2023).

## 3. Results and Discussion

Table 1 lists the 84 genes reported in [4] as being in phylostratum 10, each recorded with its HGNC symbol together with its step value, and the number of research groups (of the 13 considered) whose reported phylostratum number agreed with the modal value (here 10):

The mean for the step value of these phylostratum 10 genes is 2.743 and its standard distribution 1.240. It would not be surprising, therefore, to find that many of those in the list of 84 were incorrectly assigned to phylostratum 10, being, in reality, one or two levels higher or lower. It was thought necessary, therefore, to curate the list by conducting a BLAST search for each gene on the list to check that it did appear annotated as such in the tunicates and did not appear as such in the branchiostomatidae nor in the earlier-appearing phylostrata. Only 25 genes of the 84 survived this curation.

Inasmuch as genes with phylostratum numbers reported as being above 10 might be incorrectly listed despite actually being in phylostratum 10, we searched the literature for genes reported as being present in tunicates. If they were not already in the list of 25 curated 10 s, they were subjected to a BLAST search as just described for their presence in tunicates and not in earlier animals. In total, 59 such genes were identified and are listed together with the original 25 in Appendix A, the total number of curated tunicate-vertebrate; the number of shared orthologs is thus 84 (coincidentally the same number as in the original list). For each gene in the list, the Expect value is listed for the 2-sequence comparison between that tunicate sequence and the corresponding human gene, as well as additional information concerning that gene.

The list of 84 orthologs in Appendix A is almost certainly not a complete accounting. Many genes that the previous estimates have identified as coming from the large contribution from the fish might be revealed to be tunicate-derived orthologs if systematically studied by our hand-curation methodology.

### 3.1. The Muscle Protein Genes

#### The Proteins of Muscle

Muscular tissue can be found as early as the Porifera, where the sponges have the capacity for coordinated whole-body contractions that enable them to expel sediments [5]. Their myocytes contain a myosin homologous to the myosins of higher organisms but the myosin of the myocytes contracts and expands in response to changes in the concentration of calcium and is not ATP-dependent, nor is it activated by any neural connection [6]. The muscular system is already well-developed in the sea anemone, where muscles can be found in the tentacles as well as in the body column [7]. The sea anemone being diploblastic, its muscles are, of course, not of mesodermal origin. In the triploblastic sea urchin, a complex muscular jaw apparatus is present that can scrape and tear the animal’s food items [8]. With the emergence of the cephalochordates such as the lancelet (amphioxus sp. of the branchiostomes), muscles line the notochord to allow the active swimming behaviour of the animal. In the tunicates, swimming is further developed, and in the Appendicularians, which retain the larval form for their entire lifespan, the tail is used to sweep food particles into the oral apparatus [9].

At each stage of evolutionary development, the range of molecules involved in muscle formation is, of course, increased. These molecules include actins, the light chain and heavy chain myosins, titin, meromyosins, myosin binding proteins, troponins, and tropomyosin. In most of these classes, tunicate orthologs contribute, in some cases, decisively, as we will now illustrate.

The muscle protein orthologs are specifically denoted as such in Appendix A.

The light chain myosins have 12 representatives in the human genome with HGNC symbols beginning with MYL. Figure 2 depicts a phylogram of these proteins, rooted at the annotation of the coral ortholog of MYL6.

In addition to the coral’s MYL6, the proteins MYL2 and MYL6B also have orthologs in the corals. MYL4, MYL5, and MYL7 (in bold in the figure) have orthologs in the tunicates, while all the other proteins have orthologs either in the Echinodermata or in the Cephalochordata, leaving it to those four tunicate proteins to complete the family.

In a comprehensive analysis of the proteins in skeletal and cardiac muscles, Lindskog et al. (2015) found that MYL2 and MYL3 were present in both tissues, MYL1 only in skeletal muscle, while MYL4 and MYL7 were present only in cardiac muscle [10].

In the fourteen-membered heavy chain myosin class (having HGNC symbols beginning with MYH), the tunicate orthologs contribute five examples (Figure 3 below).

MYH14 and MYH7B (underlined) appeared only with the fish but the tunicates had contributed a substantial portion of the earlier orthologs.

Lindskog et al. (2015) found that MYH7 and MYK7b were present in both skeletal and cardiac tissues, MYH1, MHY2, MYH4, and MYH8 only in skeletal muscle, while MYH6 was only in cardiac muscle [10].

The tunicates contributed all three of the myosin binding protein C family (the MYBPCs). MYBPC1 is present in slow skeletal muscle, MYBPC2 in fast skeletal muscle, and MYBPC3 in cardiac muscle (Figure 4).

The tunicates contributed one of the three myomesin proteins (the MYOMs). The myomesins are present in the M band of striated muscles and act by stabilising the sarcomere during contraction of the muscle, acting as a molecular spring [11].

Two other proteins in the list presented by Lindskog et al. [10] are tunicate orthologs and thus appear, annotated, in Appendix A. One is synaptopodin 2 (SYNPO2), the first to appear in evolution of the two synaptopodins, which are molecules that regulate actin fibres. The second is troponin T type 1 (TNTT1), the last to appear of the three TNTTs that regulate muscle contraction in response to alterations in intracellular calcium ion concentration. The tunicate orthologs of the genes involved in muscular function form an important contribution, extending the complement of muscle genes already present in earlier animals, thus providing a basis for the radiation of such genes in the evolution of the vertebrates.

### 3.2. The Gap Junction Proteins

The human genome contains 21 Gap Junction proteins (also known as connexins). They are divided by sequence similarity into five subgroups labelled as GJAP’s through GJEP’s or with the corresponding Greek letters, α through ε. Figure 5 shows a phylogram of these proteins:

(See also [12,13] for fuller discussions of the evolution and biology of the Gap Junction proteins). Seven of the gap junction proteins are found in Appendix A of the Orthologs at the base of the Olfactores, and these are labelled accordingly in the figure. Results of BLAST searches against the tunicate genomes are listed in Appendix A.

As can be seen in Figure 5, the A, B, C, and D subgroups all have representatives from the tunicates that, in each case, may be suggested to be the progenitors of that subgroup.

The Branchiostomatidae genomes do not contain gap Junction proteins so the gap junction proteins may be considered as momentous inventions of the founders of the Olfactores clade. Gap junction proteins are transmembrane proteins that connect one cell with its neighbour. Among many other roles in the body, gap junction proteins are found in the Schwann cells, whose development in evolution required, as we have seen, the tunicates’ invention of the first gap junction proteins. These Schwann cells form a sheath that wraps around a neuron and provides the insulating myelin coating to the nerves of the bony fishes and higher organisms. Gap junction proteins connect each Schwann cell with its neighbour in the sheath and thus provide a route that enables rapid cell to cell movement of solutes from the nerve cell interior to the extracellular surface. A build-up of potassium within the nerve cell is thus mitigated and nutrient flow from the extracellular medium to the interior of the nerve is facilitated. This transport has been estimated to be a million times faster than such movement would otherwise be in the absence of the gap junction proteins. The myelin coating that the Schwann cells provide insulates the nerve and allows electrical signals to pass from the interior to the exterior only where the Schwann cell coating is absent at the internodes. The electrical signal thus jumps (“saltatory movement”) from internode to internode, speeding up manyfold the rate of signal transmission. For example, whereas unmyelinated axon conduction velocities range from about 0.5 to 10 m/s, myelinated axons can conduct at velocities up to 150 m/s. [14]. In a giraffe, a signal that takes some 60 milliseconds to travel from the foot to the brain [15] will instead take some 6 s. It is to be noted that the cyclostomes such as the lamprey do not produce myelin, although they do possess Schwann cells [16]. Indeed, BLAST searches of the lamprey genome for some of the genes that a GeneCards search suggested as being involved in myelin production (MPZ, MBP, PMP2, and MAG) proved fruitless. Thus, myelin is an invention of the jawed fish.

Some myelin-associated genes (MYRF, PLP1, and PMP2) are present in the Tunicate genome (see also [17]) and, as we have seen, seven Gap Junction genes are present in the tunicates. Their function in the tunicates does not seem to be known, but a search of ANISEED, the ascidian database (see Methods), for the Gap Junction gene GJC1 returned:

REG00001500 (Cirobu.REG.KhC2.5771334-5773564|Msx)

In which, for this gene, a “CRM (cis-regulatory module) is active in the region of the primordial pharynx, in the sensory vesicle, and in the neck connecting the sensory vesicle, in the visceral ganglion, and in the caudal muscles. At earlier stages, it is active along the neural tube in the tailbud, along the neural folds in the neurula, in the precursors of sensory organs, in the ventral epidermis in both stages, and also in the mesoderm that will give rise to the muscles”. Presumably this gene and its congeners were co-opted by the ancestor of the Schwann cell to enable the formation of the myelinated nerves of the jawed fish and their descendants.

We searched the genome of the Branchiostomatidae for a gene that could be the ancestor of the gap junction genes of the tunicates. We found that GJAP8 had, as top hit with the Branchiostomatidae, a gene denoted as the “predicted Titin homologue” annotated as XP_019638343.1. This had an Expect Value against GJAP8 of 9 × 10^−11^. The protein had a length of 1894 amino acids. Figure 6 depicts the alignment of this protein against the seven tunicate Gap Junction genes; Figure 6A shows the alignment against the whole Titin sequence while Figure 6B shows the alignment against the 891 portion of the sequence that spans those of the shorter GAP Junction sequences.

Figure 7 shows a phylogram with the 891-long sequence of the Titin homologue and these seven tunicate Gap Junction proteins. The data suggest that a portion of a member of the Branchiostomatidae titins might have evolved to produce the Gap Junction proteins of the Olfactores.

### 3.3. The Cadherins

The human genome contains 23 cadherin (CDH) genes, with HGNC symbols *CDH1* through *CDH26* (three are absent). Of these twenty-three, seven (*CDH1*, *CDH2*, *CDH7*, *CDH8*, *CDH11*, *CDH16*, and *CDH18*) are found as orthologs in the Tunicates while another (*CDH23*) is an ortholog in the Branchiostomatidae. The twenty-three CDH genes are divided into two types, I and II. These *CDH* genes code for correspondingly-named membrane-bound CDH proteins, each containing multiple cadherin repeats of some 110 amino-acids. The Type II cadherins are missing a long sequence of amino acids in the N-terminal region. Refer to Appendix A.

The upper two rows in Appendix A depict cadherins of type I, while the lower rows depict those of type II. In each case, an ortholog found in the tunicates is marked with an asterisk. Appendix A depicts the full sequences while Appendix A shows the N-terminal sequences expanded. A cadherin from *Homo sapiens* and its tunicate ortholog show very close alignment. The green-shaded region of the sequences in Appendix A marks the membrane-embedded portion of the proteins.

A phylogram of all the cadherins from *H. sapiens* is presented in Figure 8.

As can be seen, each of the two cadherin types contains proteins (labelled “tunicate”) that have orthologs in the tunicates and might perhaps be considered as founder members of that type. The single cadherin ortholog that derives from the Branchiostomatidae (designated by “lancelet”) lies separate from the cadherins in the two types (I and II).

The cadherins are well-named, being proteins that cause cells to adhere to one another under the influence of calcium (Ca). An x-ray crystallographic study of cadherins by Patel et al. (2005) showed that it was largely the N-terminal sequences of the molecules that determined the binding between two cadherins [18]. The N-terminal sequence of one molecule formed an anchor that fitted into a pocket in the second molecule, whose N-terminal sequence in turn inserted into a pocket on the first molecule. The process is called “swapping” of the two N-terminal sequences, although of course no chemical bonds are broken in this swapping. The different structures of the anchor (see the N-terminal sequences of the two types of cadherins in Figure 6 and pocket between type I and type II cadherins render the two types incompatible with each other. This incompatibility is of profound significance in embryological development. Cells from an epithelial sheet of cells held together by cadherins of one type (say Type I) will detach from the sheet if they are temporally programmed to express cadherins of Type II. This process, accompanied by the programmed lowering of expression of Type I cells, is of crucial significance in the formation and detachment of the neural crest [19].

Table 2 lists some frequently studied cadherins with the commonly used names that they are referred to in the literature, the tissue in the body where each protein is usually found, and the HGNC symbol for each protein.

Cells bearing cadherins of one name will readily form homotopic interactions, but less frequently heterotopic combinations, with other cadherins, and this leads to sorting-out of different cell types in embryological development.

In the animal body, the cadherins exert their adhesion function in many epithelial tissues but also, importantly, in neural tissue and the formation and detachment of the neural crest. Cells from the neural crest migrate throughout the embryo, giving rise to an array of cell types that characterize the vertebrate clade, including the peripheral sensory nervous system and most of the craniofacial skeleton.

In the cephalochordate Amphioxus, two cadherin paralogs are found [20]; one is expressed in the mesoderm and the other in the ectoderm, but both are found expressed in neural tissue, at different developmental periods. These two paralogs are, of course, of the same type and can only demonstrate adhesion and not repulsion. In Amphioxus, there are no cells that have been identified as homologous to neural crest [21]. The tunicates possess cadherins of both types so that both cohesive and repulsive interactions are now possible. In the tunicates [22], a CDH5-like protein is expressed at the tailbud stage in the nerve cord and in the peripheral neurons of the tail, while the CDH7-like protein is widely expressed in the epidermis and also in the nerve cord, the sensory vesicle, and the visceral ganglion. Neural crest-like cells are indeed found in the tunicates as migratory cells that produce pigment (as do vertebrate cells of neural crest origin) and, in addition, express many of the genes that regulate the development of vertebrate neural crest [23].

The role of the cadherins expanded greatly as the number of cadherins themselves expanded and diversified during the evolution of the vertebrates, with multiple roles in embryological morphogenesis [24].

An additional set of proteins involved in cell adhesion and, in particular, in cortical development and synapse formation [25] are the FLRTs (Fibronectin Leucine Rich Transmembrane proteins), of which FLRT2 and FLRT3 were tunicate innovations and which, in their developmental role, interact with latrophilin (HGNC symbol ADGRL3), also first found in the tunicates. Latrophilin and the FLRTs also interact with teneurin proteins (the TNFMs) to form trimeric complexes, important during synapse formation and in guiding the migration of young neurons [26].

### 3.4. The Claudins

The human genome contains 24 claudin genes symbolised by *CLDN*, enumerated up to *CLDN34* with some absent members. These genes, of course, code for the correspondingly named proteins. The origin of the world claudin [27] is from the Latin *claudere*, meaning “to close”, very appropriate since their function is to form the Tight Junctions between adjacent cells in an epithelium or endothelium [28]. These Tight Junctions ensure that passage through them, from one side of an epithelium to the other, is tightly controlled in a selective permeability, with the different claudins having different selectivities [28]. Orthologs of the claudins are found in the tunicates (three of them) while another is found in the earlier-appearing Branchiostomatidae, but not in the even earlier Echinodermta. Figure 9 depicts a phylogram of the human claudin proteins with the earliest-appearing genes designated.

The figure suggests the close similarity of CLDN7, the first claudin to appear (in the Bran-chiostomatidae) to two of the next-appearing (in the tunicates): CLDN1 and CLDN19; the third tunicate claudin, CLDN18, has close similarity to the lamprey claudin CLDN10 and the two shark claudins CLDN11 and CLDN15.

Figure 10 shows a comparison of the sequence of the CLDN18 orthologs from the tunicate *Ciona intestinalis* and the lamprey *Petromyzin marinus*, that of the lamprey being shown as the lower sequence of the two in each row. The two sequences, analysed in a BLAST search against one comparison, returned an Expect Value of 2 × 10^−9^.

Defects in claudin genes are associated with congenital deafness in children. Studying the expression of claudin genes in the Zebrafish, Kollmar et al. (2001) found claudins in the otic and lateral-line placodes of the fish [29]. They suggested, on the basis of their findings, that claudins could have an additional role in vertebrate morphogenesis. Perhaps the role of the early claudins in tunicates might relate to a role in tunicate morphogenesis in addition to their function as aiding to form barriers between the external and internal environments, as indicated by electron microscopy studies.

### 3.5. The Ephrons and Ephrins

The human genome contains 14 ephron genes, symbolised by *EPHA1* to *EPHA10* (*EPHA9* is absent) and *EPHB* to *EPHB4* [30]. These code for corresponding membrane-bound proteins. Table 3 includes these genes together with the animal species where their latest shared orthologs appear.

Figure 11A displays a phylogram of the relation between the proteins of the A series, the phylogram being rooted at EPHA5, while Figure 11B displays the corresponding phylogram for the proteins of the B series, the phylogram being rooted at EPHB2. The ephron proteins are membrane-bound receptors for the eight ligand Ephrin proteins, symbolized by EFNA1 through to EFNA5 and EFNB1 through EFNB3, coded for by the corresponding *EFN* genes [30].

Table 3 also includes these genes together with the animal species where their latest shared orthologs appear.

Figure 12 displays a phylogram of the relation between these proteinss, the A and B series being clearly separated into their two subgroups.

Since both the ephrons and the ephrins are membrane-bound, their interactions are only cell to cell. The proteins of the A series of the receptor ephrons bind to the proteins of the A series of the ligand ephrins and repel the proteins of the B series of the ephrins. B series ephrons bind B series ephrins and repel those of the A series. These attractive and repulsive interactions between cells determine many morphogenetic processes in animal development [31]. Repulsive effects determine the boundaries between different cell types. These repulsive responses are likely to involve depolymerization of the actin cytoskeleton, leading to the collapse of the filopedia and retraction of the cells involved. Actin-cytoskeletal interactions are the probable base of the attractive responses. A combination of repulsive and attractive effects guides the movement of nerve cells during the embryonic formation of the brain and ensures appropriate movement of cells in the intestinal crypt, retaining the Paneth at the base of the crypt while guiding the movement of cells of the crypt towards the lumen of the intestine.

The ancestor of the Olfactores contributed none of the 14 Ephron genes but as many as 4 of the 8 Ephrin genes, a substantial contribution. Two Ephron genes (*EPHA4* and *EPHA2*) were already present in the Cnidarians as well as one ephrin gene (*EFNB1*). In the tunicates themselves, these genes are involved in (among other processes) neural induction [32], endoderm invagination [33], asymmetric cell divisions in the vegetal hemisphere of the developing brain [34], delimiting the number of pigment cells in the central nervous system [35], and neural tube patterning [36].

### 3.6. The MAGE Genes

There are some 40 *MAGE* genes (the name derives from Melanoma Antigen Gene) in the human genome, divided into 2 subfamilies, Type I and Type II, according to their sequence and chromosome location [37]. The Type II family can again be subdivided according to their ancestry: those originating from an ortholog in the tunicates, descended from the tunicate gene Cirobu.g00007177 (also known as “melanoma-associated antigen D2 isoform X2 [*C. intestinalis*]”) and being named in the form *MAGEDx*, and those originating from an ortholog in the branchiostomata, most being named *MAGEEx*. The phylogram in Figure 13 shows how the Type II MAGE proteins divide into these two subfamilies, together with a separate protein NDN, necdin, also a member of the MAGE family.

The biological roles of the tunicate-contributed MAGE proteins are tabulated in Appendix A, and include a role in cell adhesion, in the regulation of ubiquitin-protein ligases, and response to stress.

### 3.7. The Crystallins

#### The Gamma Crystallins

The human genome contains 18 crystallin proteins. They are divided by sequence similarity into three subgroups, the α, β, and γ crystallins and, interacting together, they form the transparent lens of the eye. The six gamma crystallins are descended from a single ortholog in the Tunicates, namely the gene designated in the ANISEED database (see Methods) with the unique ID of Cirobu.g00014781. This gene can appear in BLAST searches of the Tunicates as one of three annotated sequences, two from *C. intestinalis* (XP_002126888.1 (named as gamma-crystallin S) and 2BV2_A (Named as Chain A, *C.* betagamma-crystallin)) and one from *Styela clava* (XP_039266482.1 (named as gamma-crystallin N-A-like)). In what follows, the sequence 2BV2_A has been used. The HGNC symbols for the six gamma crystallins of *H. sapiens* are CRYGA, CRYGB, CRYGC, CRYGD, CRYGN, and CRYGS. The three crystallin genes *CRYBG1*, *CRYBG2*, and *CRYBG3*, coding for proteins that also present in the vertebrate lens, but some ten-fold longer than the gamma crystallins, will not be discussed here. Appendix A, part A depicts a COBALT-based alignment of these proteins compared with the six gamma crystallins and the single tunicate crystallin, while part B is a phylogram from these data. The gamma crystallins and the CRYBG proteins clearly separate into two families. The *CRYBG* genes are not found in the genome of the tunicates, although *CRYBG2* is present in the Branchiostomatidae. These genes have curiously named synonyms of the form *AIMx*, where *AIM* stands for Absent In Melanoma [38].

Figure 14 depicts a COBALT-derived comparison of the six gamma crystallins together with the *C.* crystallin.

Figure 15 depicts the Phylogram that COBALT produced from the data, here rooted at the *C.* crystallin.

Note in Figure 14 that the tunicate crystallin is almost exactly half the length of the crystallins of *H. sapiens*, suggesting that the tunicate gene was doubled as it evolved into its vertebrate descendants. This suggestion is confirmed if the sequences of the tunicate crystallin and two vertebrate crystallins are compared, as in the dotplots of Figure 16:

The 83 residue sequence of the tunicate protein is repeated in the second half of both vertebrate proteins with a minor modification: a small break in the second half of the sequence of CRYGN (part A of Figure 16) and in the first half of the sequence for CRYGS (part B of Figure 16). This difference between the two plots suggests that the two vertebrate proteins arose by two independent gene duplications, compatible with the positions of these two vertebrate proteins in the phylogram of Figure 15. It would appear, from the phylogram, that the further speciation of the crystallins originated with CRYGS. These two crystallin proteins, CRYGN and CRYGS, are both found in our table of the orthologs at the base of the Olfactores since they appear to have evolved independently.

It was mentioned at the beginning of this section that, in vertebrates, the crystallins are found in the lens of the eye. In a tunicate, however, the crystallin protein is not found where one might expect it to be: in the ocellus or eyespot of the organism. Rather the crystallin is found in the palp, an organ in the snout of the larva of the sessile tunicates whose function is to provide, together with lectins and other components, a mucilage that fixes the tunicate to the seabed [44]. That the tunicate crystallin is not expressed in the organism’s eyespot suggested to Horie et al. (2008) “that the lens of the *C.* pigmented ocellus is not homologous with that of the vertebrate eye” [45].

The tunicate crystallin can bind calcium [46], thereby stabilising the protein against temperature denaturation. Vertebrate crystallins have lost these calcium-binding sites. Figure 17 shows a comparison of part of the amino-acid sequence of the lamprey’s CRYGN (upper row) and the tunicate crystallin (lower row). The wide arrows point to the sites at which calcium binds in the tunicate protein [46].

These mutations and the duplication of the tunicate’s sequence led to the dramatic evolution of the protein from a mucilage in the tunicates into the vertebrate proteins that confer upon the lens its transparency and hence the ability of the vertebrates to use their visual system in efficient predation.

### 3.8. The Distal-Less Genes

The human genome contains six members of the distal-less family with HGNC symbols *DLX1* through *DLX6*. Of these, *DLX1*, *DLX4*, and *DLX6* first appear in the Cnidaria, with *DLX 1* and *4* present in corals and *DLX6* in a sea anemone. *DLX5* first appears in the Echinodermata, in a sea urchin, while *DLX2* and *DLX3* were the contribution of the tunicates (Figure 18)

A multiple alignment of these genes is shown as Figure 19:

A member of the gene family was first noticed in the fruit fly, *D. melanogaster*, as the gene *Dll* when a mutant lacking the distal ends of the limbs was found (reviewed in [47]), and the gene was postulated to control pattern formation along the proximal-distal axis of the limbs. A similar role in controlling pattern formation along the limb proximal-distal axis is found in the Crustacea [48]. The phenomenon of an absence of distal regions of the limbs is also found in mice, where targeted inactivation of the genes *DLX5* and *DLX6* led to severe malformation of the distal portion of the limbs. The human disease Split-hand/split-foot malformation (SHFM) is a human limb defect characterized by missing digits and fusion of remaining digits. Ullah et al. (2016) showed that a mutation in *DLX6* is associated with the type I form of SHFM [49].

In the mice, disruption of *DLX* genes led to gross malformation of the craniofacial skeleton. These craniofacial defects, associated with distal-less gene mutations in the mouse, suggest that distal-less controls more than simply proximal-distal pattern information. Indeed, *Dll*, in its first appearance in Hydra (where, of course, no proximal-distal axis is present), is expressed in the head, battery, and the stem-like peduncle [50]. In Hydra, potent stinging cells are grouped with other neurons in what are called “battery complexes” on the hydra’s tentacles. Indeed, the mutation in the Drosophila *Dll* gene is also associated with defects in the antenna of the fly, a sensory organ. In the sea urchin (which too has no proximal-distal axis), ectodermal expression of distal-less is evident at the tips of spines but is not associated with other skeletal elements [51]. In the myriapod *Glomeris marginata*, the distal-less gene is expressed in the limb appendages but also, and to a large extent, in the mouth parts of the organism [52]. In the tunicate *C. intestinalis*, DLX3 is expressed in the atrial siphon and in addition in the adhesive organ, which is located in the head, at the anterior end of the body, and is used by the larvae at the beginning of metamorphosis to attach to a solid substrate [53]. Finally, in Amphioxus (the lancelet *Branchiostoma floridae)*, a context where the proximal-distal axis is again not relevant, Holland et al. (1996) [54] showed that the animal’s single *Dll* gene is expressed in the anterior three fourths of the cerebral vesicle and is implicated in the “establishment of the dorsoventral axis, specification of migratory epidermal cells early in neurulation and the specification of forebrain”. They add the intriguing suggestion that “Such a multiplicity of Distal-less functions probably represents an ancestral chordate condition and, during craniate evolution, when this gene diversified into a family of six members, the original functions evidently tended to be parcelled out among the descendant genes”. Interestingly, in the parallel bilaterian evolutionary path to the insects, as we saw in the case of the Myriapod, the ancestral *DLX* gene again took on the function of regulating pattern formation in the proximal-distal axis and as well as specifying head development and aspects of sensory perception. Thus, the *DLX* genes have evolved to be interpreters that express their function by being directed to various locations in the body. In accordance with signalling systems present at this location, the *DLX* genes drive and direct the expression of genes that act together to form the particular organ or organs there found, noteworthy in the development of the limbs and the craniofacial skeleton.

### 3.9. DAVID Analysis of Individual Orthologs Not Discussed in Previous Subsections

In the list of the 84 tunicate orthologs assembled in Appendix A, 36 have not yet been covered in the text so far. To tease out what might be their role in the tunicate lifestyle and perhaps in that of the vertebrates as well, we subjected the 36 to an analysis using the DAVID program (see Methods). As it describes itself, DAVID is “The Database for Annotation, Visualization and Integrated Discovery (DAVID) [that] provides a comprehensive set of functional annotation tools for investigators to understand the biological meaning behind large lists of genes”. We applied the functional annotation clustering tool of DAVID (which attempts to find common themes in groups of genes selected from a submitted list). Table 4 is extracted from the full output of the DAVID clustering.

Of these listed clusters, cluster 2 and cluster 4 continue the emphasis on adhesion between cells that was discussed previously in the sections on the cadherins and the ephrons and ephrins, while cluster 6 extends the list of genes concerned with muscular contraction. Cluster 7 contains genes concerned with embryological development while cluster 8 concerns the important zinc-finger genes with their widespread roles in gene regulation.

Information abstracted from the summaries in the GeneCards database (see Methods) is listed for each of these 36 genes in Appendix A, together with all the orthologs that are assembled in that table.

### 3.10. Tunicate Orthologs That Are Non-Coding Genes

In their 2019 paper, Litman and Stein [4] estimated the ages of every one of the 19,651 protein-coding genes but succeeded in this endeavor for only 5981 of the 16,528 non-protein-coding genes of the human genome. In total, 23 of these 59,281 were contributed by the tunicate clade. Analysing these using the orthologs platform of GeneCards (see Methods) showed all 23 to be Ascidian orthologs to many vertebrate genomes, none of them having a representative in any animal lower than the Ascidians. The 23 are listed in Appendix A, together with excerpts of the annotations found using DAVID’s Functional Analysis program (see Methods). Many of the microRNAs are listed as being implicated in the development of the muscular, neuronal, and vascular tissues, so important in the predatory behavior of vertebrates.

## 4. Conclusions

In the list that partitioned the 19,653 protein-coding genes of the human gene into their appropriate phylostratum level, Litman and Stein (2018) found, as stated earlier, that 84 of these genes fell into phylostratum 10, associated with the tunicates [4]. Thus, these tunicate-shared genes, which stand at the base of the Olfactores clade, are a very minor fraction of the genes of the human genome. This tunicate contribution was preceded by a large contribution of genes from the Branchiostomatidae (406) and was followed by an even larger number from the fishes (3650). The hand-curated list of phylostratum 10 genes that are listed in Appendix A of this paper again contains only 84 genes. That there are orthologs that are found shared with the vertebrates in the tunicates, and yet are not present in the Branchiostomatidae, accords with the suggestion of [2] Delsuc 2008 that the tunicates and vertebrates are sister clades of the Olfactores and succeeded the Branchiostomatidae in evolutionary time.

We were intrigued that this small number of 84 genes at the base of the Olfactores could open the vast evolutionary changes that led, on the one hand, to the tunicates themselves and, on the other, to the vertebrates. One must add in here the 23 non-coding ortholog genes that first appeared with the tunicates (Appendix A) and are heavily implicated in the evolutionary development of vertebrate muscle, nerve, and vascular system, with MIR133B, in particular, being closely involved in all three of these systems.

Perhaps the most important contribution of the tunicates was the evolution of a second type of cadherin. This, a Type II cadherin, had the property of detaching the cell containing that cadherin from cells that expressed the Type I class. The set of such Type II cadherins could now detach and move away from their Type I neighbours, a process that would eventually evolve into the formation of the neural crest. Cells from the neural crest migrate throughout the embryo giving rise to an array of cell types that characterize the vertebrate clade, including the peripheral sensory nervous system and most of the craniofacial skeleton. Indeed, the neural crest can be considered as the fourth germ layer, providing a wide range of possibilities for further evolutionary invention [55]. Controlled interactions between cadherins are of prime importance in the formation of the skin appendage placodes, the cell thickenings, and budding from which arise the teeth, scales, hair, feathers, and mammary glands of the vertebrates.

A second important contribution was the broad development of the muscle and nerve protein toolkits. The tunicates contributed all three of the MYBPC genes that populate the M-band of the sarcomere, leading to a great improvement in the functioning of muscle and the tunicates contributed numerous genes to the other classes of muscle proteins. The tunicates “invented” the first gap junction genes, so important in the Schwann cells and the development of rapidly transmitting myelinated axons. The crystalline genes of the lens of the eye are again of tunicate origin being found in the tunicate palp’s mucilage that enables the larval tunicate to attach to the sea bottom. These developments in mobility and vision provided the basis for the development of the efficient predatory capabilities of the Vertebrata. The evolutionary change from the lifestyle of the Branchiostomatidae to the shared portion of the lifestyles of the Tunicata and vertebrata is perhaps not so great a leap and the 84 orthologs that lie at the base of the Olfactores clade were clearly adequate to bridge this gap.

## Figures and Tables

**Figure 1 genes-15-00657-f001:**
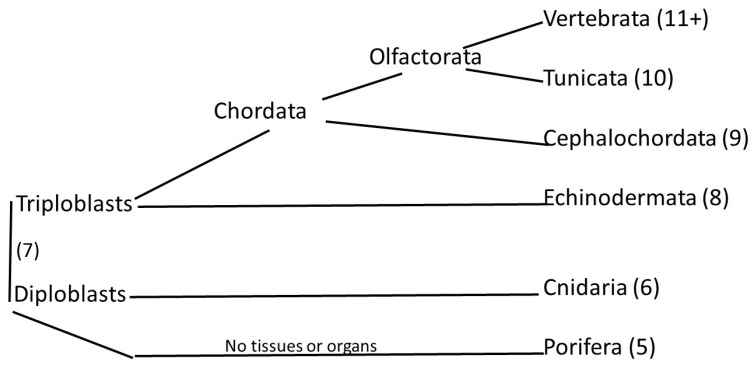
Cladistic diagram of animal evolution with the tunicates as the sister clade of the vertebrates, together forming the Olfactores clade [2].

**Figure 2 genes-15-00657-f002:**
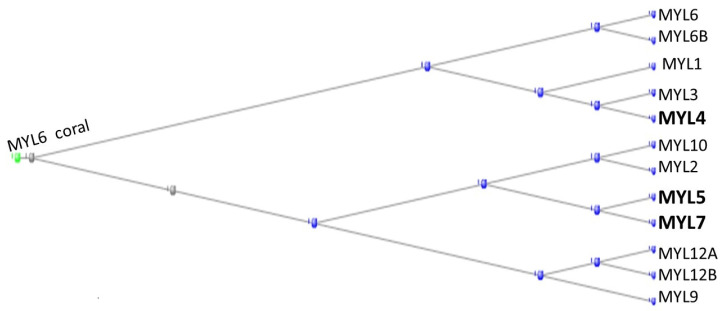
Phylogram of the MYL proteins, rooted at the Coral ortholog of MYL6. The orthologs contributed by the tunicates are shown in bold type.

**Figure 3 genes-15-00657-f003:**
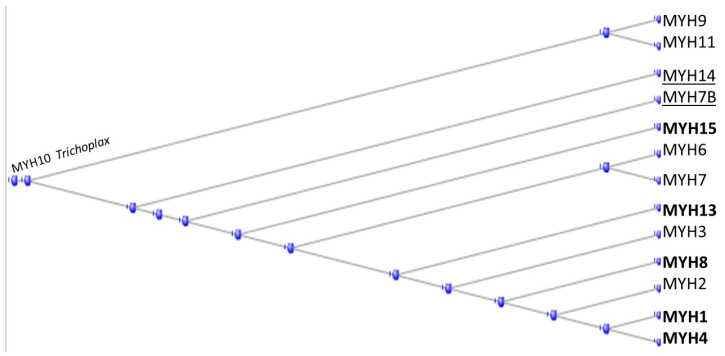
Phylogram of the MYH proteins, rooted at the annotation of the Trichoplax ortholog of MYH10. Tunicate orthologs in bold type. Fish orthologs underlined.

**Figure 4 genes-15-00657-f004:**
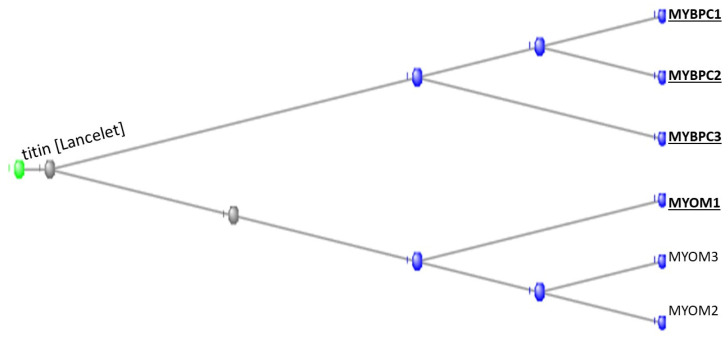
Phylogram of the MYBPC and MYOM proteins rooted at the Lancelet ortholog of the giant muscle protein Titin, which appeared with the Branchiostomata. The orthologs contributed by the tunicates are shown in bold type and underlined.

**Figure 5 genes-15-00657-f005:**
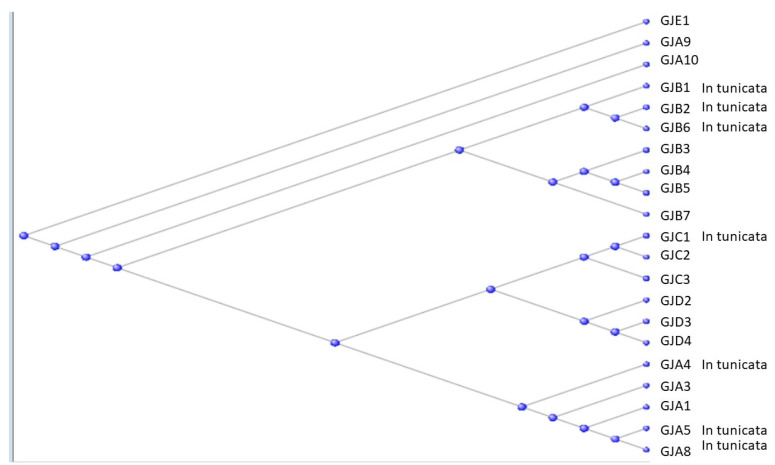
Phylogram of the Gap Junction proteins. Those found with orthologs in the Tunicata are labelled on the figure.

**Figure 6 genes-15-00657-f006:**
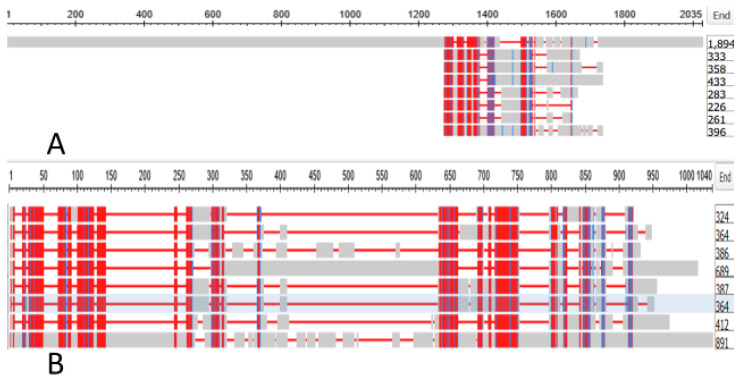
(**A**) depicts the alignment of XP_019638343.1, the annotation for the sequence of the titin homolog of *Branchiostomata belcheri* against the seven tunicate Gap Junction genes listed in Appendix A; (**B**) shows the alignment against the 891 portion of the sequence that spans those of the shorter GAP Junction sequences.

**Figure 7 genes-15-00657-f007:**
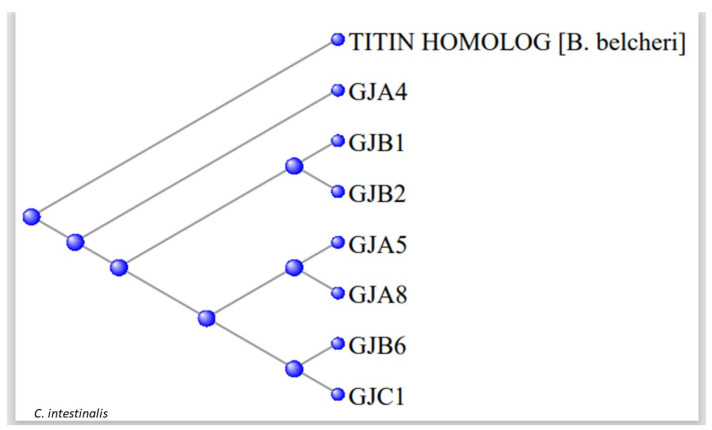
Phylogram of the seven tunicate Gap Junction proteins rooted at the 891-long sequence of the Titin *Branchiostomata belcheri* homolog.

**Figure 8 genes-15-00657-f008:**
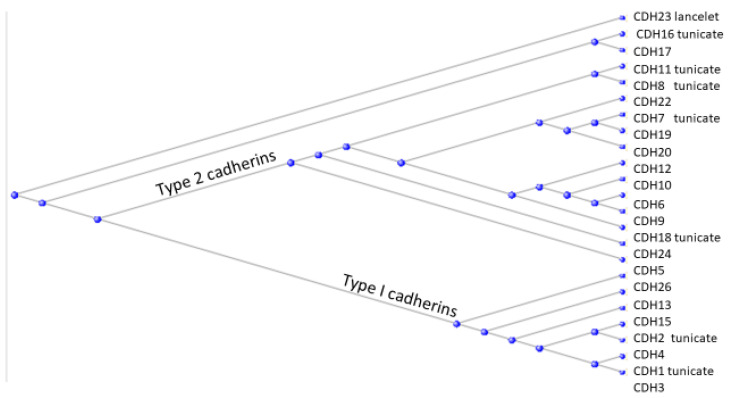
Phylogram of the cadherin proteins of *H. sapiens*. Those with tunicate orthologs are indicated as such on the figure. The single cadherin with an ortholog in the Branchiostomatidae is indicated as “lancelet”.

**Figure 9 genes-15-00657-f009:**
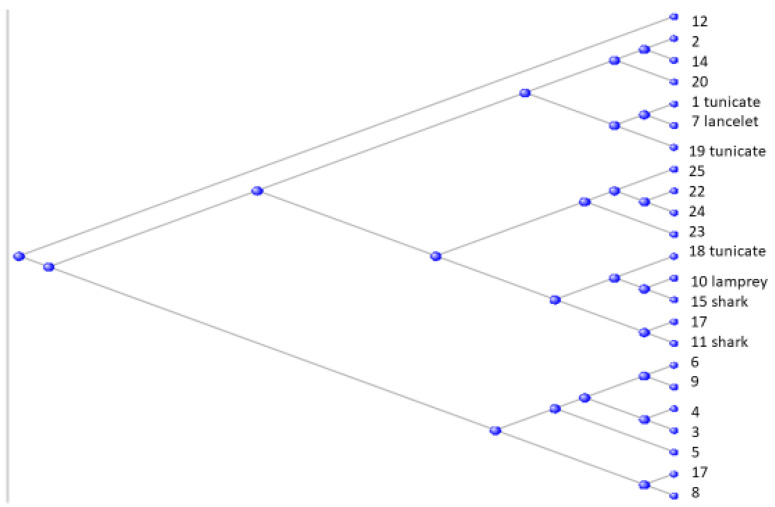
Unrooted phylogram of the human claudin proteins with the three tunicate-shared orthologs indicated as such, the single Branchiostomatidae-shared ortholog indicated as “lancelet”, the single lamprey-shared, and the two shark-shared orthologs indicated as such.

**Figure 10 genes-15-00657-f010:**
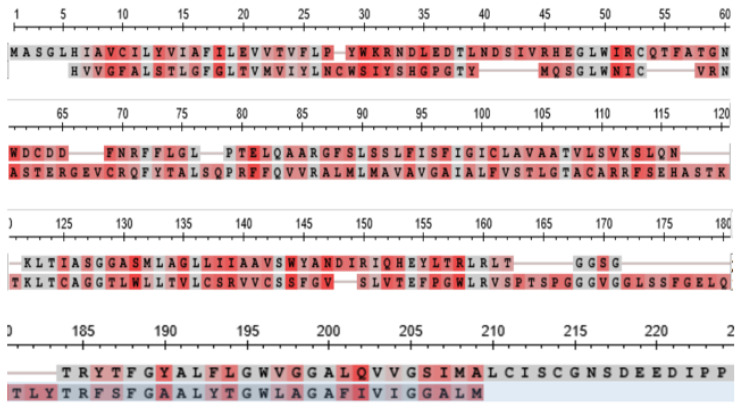
Comparison of the sequence of the CLDN18 orthologs from the tunicate *C. intestinalis* and the lamprey *P. marinus.* The sequence of the lamprey ortholog is shown as the lower sequence of the two in each row.

**Figure 11 genes-15-00657-f011:**
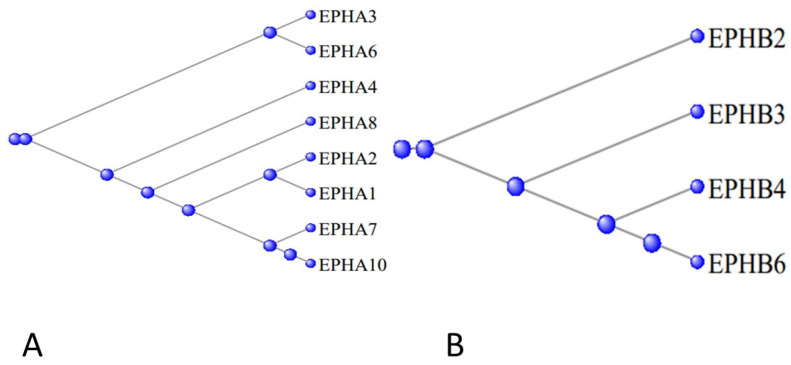
(**A**)—Phylogram of the relation between the proteins of the EPHA series, the phylogram being rooted at EPHA5. (**B**)—the corresponding phylogram for the genes of the EPHB series, the phylogram being rooted at EPHB2.

**Figure 12 genes-15-00657-f012:**
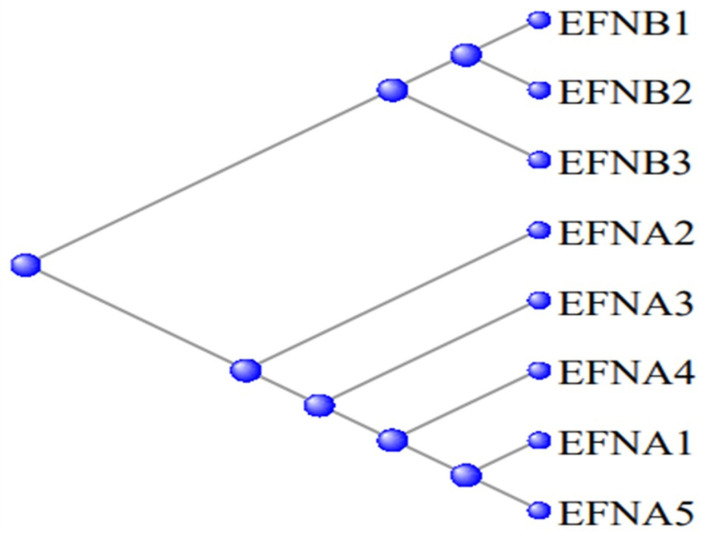
Phylogram of the relation between the Ephrin proteins—the EFNA and EFNB series separate into their two subgroups.

**Figure 13 genes-15-00657-f013:**
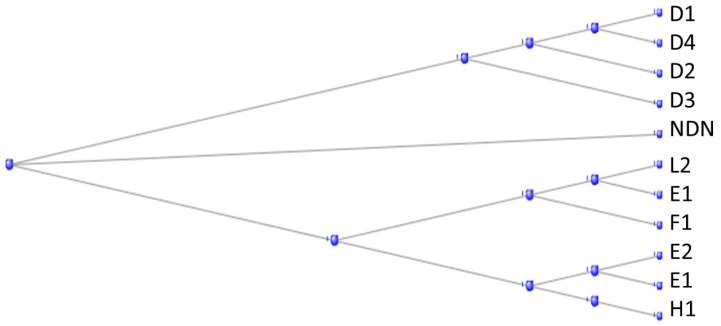
Phylogram of the Type II MAGE proteins, together with a separate protein NDN, necdin, also a member of the MAGE family.

**Figure 14 genes-15-00657-f014:**
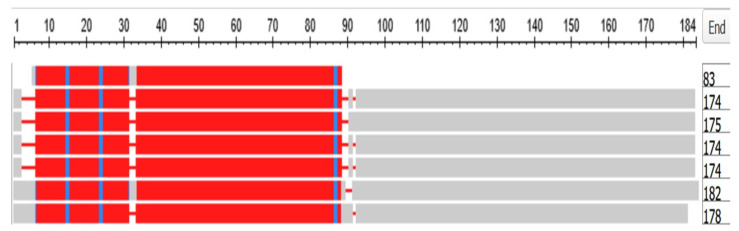
Multiple alignment of the six gamma crystallins, compared with the single crystallin of the tunicates. The figure was generated by the COBALT program of the National Library of Medicine (see Methods). From top to bottom, the proteins depicted are:Chain A, betagamma-crystallin of *C. intestinalis*, CRYGA, CRYGB, CRYGC, CRYGD, CRYGN, and CRYGS of *H. sapiens*. The length of each sequence in amino acids is listed under the heading END.

**Figure 15 genes-15-00657-f015:**
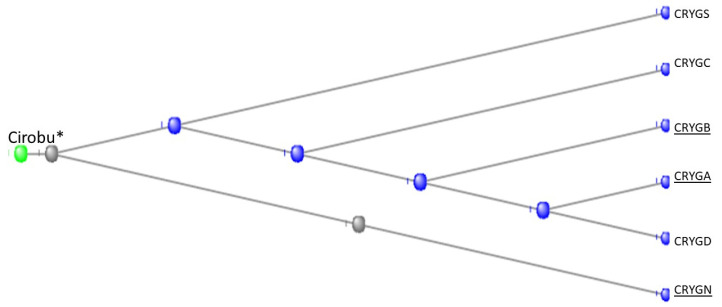
Phylogram of the six gamma crystallins together with the Tunicate crystallin. The figure was generated by the COBALT program of the National Library of Medicine (see Methods), being rooted at the Tunicate crystallin. (see also [39,40,41,42,43] for fuller discussions of the evolution and biology of the lens and of the crystallin proteins).

**Figure 16 genes-15-00657-f016:**
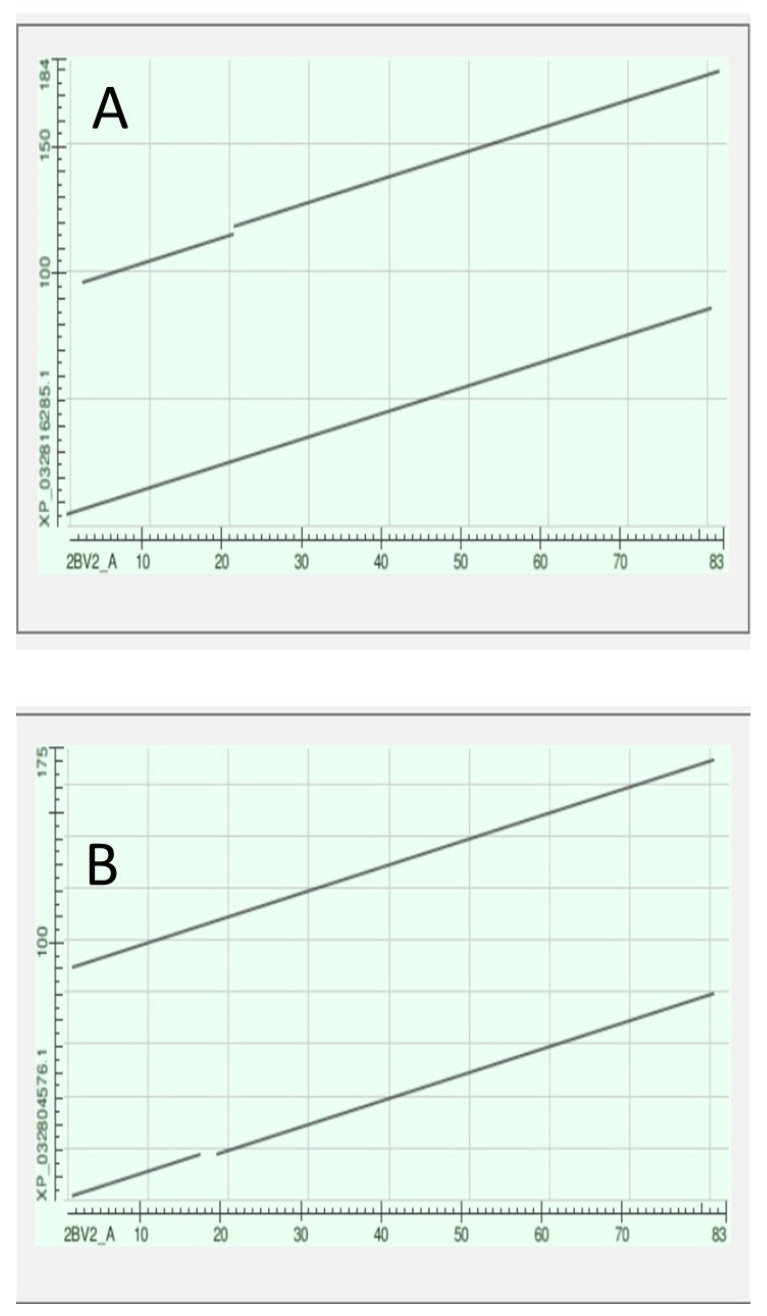
Dotplots produced by the BLAST program (see Methods) with the tunicate sequence on the x-axis and, on the y-axis, in (**A**) the CRYGN and in (**B**) the CRYGS of the lamprey *Petromyzon marinus*.

**Figure 17 genes-15-00657-f017:**
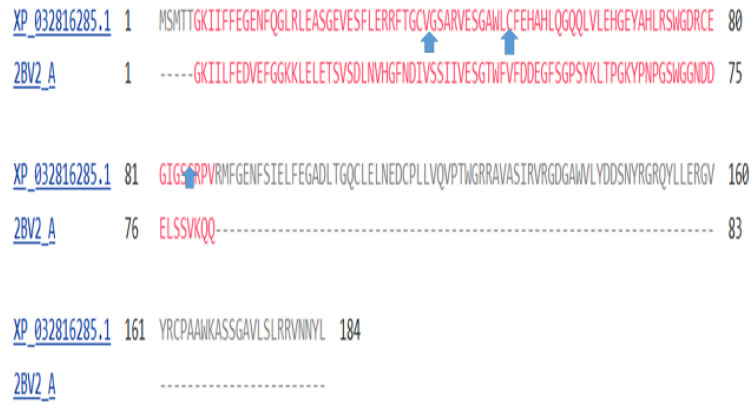
Alignment of the amino-acid sequences of the lamprey CRYGN protein and the tunicate crystallin. The wide arrows show the mutations that occurred between the lamprey and tunicate in the sites that bind calcium in the latter organism (sequence data from [46]).

**Figure 18 genes-15-00657-f018:**
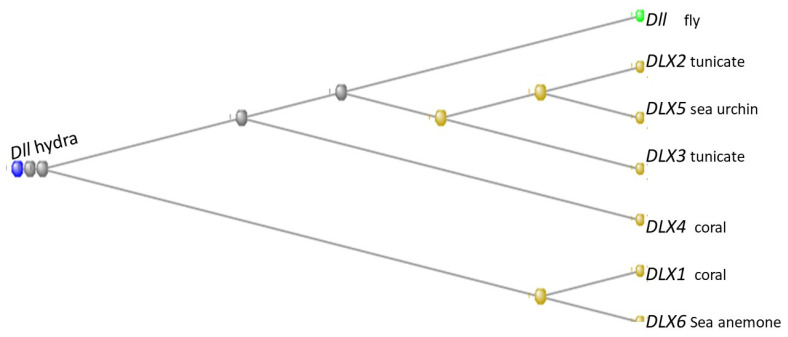
Phylogram of the six *DLX* genes of the human genome together with the gene Dll of *Drosophila melanogaster*, rooted at the *Dll* gene of *Hydra vulgaris.* The organisms in which the orthologs with the human gene first appeared are indicated next to each gene. The phylogram was built using the COBALT program (see Methods).

**Figure 19 genes-15-00657-f019:**
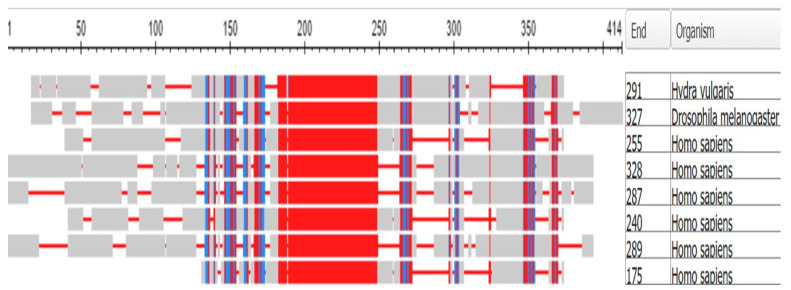
Multiple alignment of the six DLX proteins of the human genome together with the gene Dll of *Drosophila melanogaster*. The rows from the top: Dll of *Hydra*, Dll of the fly, and then DLX 2, 5, 3, 1, 6, and 4 of *H. sapiens*. The alignment was made using the COBALT program (see Methods). The number of amino-acid residues in each protein is listed under “End”.

**Table 1 genes-15-00657-t001:** Orthologs from Phylostratum 10: Consortium’s Spread as steps (see text).

HGNC	Steps	Equal to Mode	HGNC	Steps	Equal to Mode
*ADGRL3*	3.15	4	*MGAT4D*	4.08	4
*ANKRD16*	4.77	4	*MSMP*	1.15	6
*ANKRD42*	2.38	8	*MUM1*	2.10	5
*ARHGEF38*	2.67	4	*MYBPC3*	2.08	6
*ASB2*	3.92	5	*MYL5*	5.01	3
*ATRAID*	0.46	11	*NEBL*	2.16	6
*BCL6B*	2.95	4	*NECTIN3*	1.85	4
*C18ORF21*	0.62	9	*NEXN*	2.46	5
*CALML6*	2.47	8	*NPC1L1*	4.85	5
*CASR*	1.92	7	*OLFML2A*	1.54	5
*CATIP*	3.18	4	*OTULIN*	1.55	7
*CBLN1*	1.38	4	*PDX1*	2.77	4
*CCDC78*	2.62	4	*PHF21A*	2.46	4
*CEP63*	3.40	5	*PODN*	3.46	4
*CEP83*	2.46	7	*PRR14*	2.27	6
*CKAP2*	2.54	5	*RBM14*	3.61	3
*CLDN1*	0.83	7	*RHBDL2*	4.54	4
*CLDN18*	1.25	5	*RNF4*	4.85	3
*CLDN19*	0.92	7	*SCLT1*	3.18	6
*CLDN7*	1.23	6	*SERINC5*	4.38	5
*EFNA1*	1.92	4	*SLC35G2*	3.31	4
*FABP2*	1.00	8	*SLC35G3*	4.86	3
*FAM155B*	1.54	5	*SLC35G4*	5.13	3
*FAM187A*	1.23	6	*SLC35G5*	5.50	3
*FAM3D*	2.38	4	*SLC35G6*	4.25	4
*FBXO24*	2.23	6	*SLC6A14*	3.54	3
*FOXH1*	2.95	5	*SLC6A6*	3.46	4
*GJA10*	2.02	5	*SNRNP48*	2.31	8
*GJA3*	1.16	7	*SPATA21*	4.50	3
*GJC2*	2.38	6	*STXBP6*	1.77	8
*GNA15*	4.00	3	*TGFB1*	2.15	4
*GTF2IRD2*	3.00	3	*TGFB2*	1.54	6
*GTF2IRD2B*	4.30	3	*TLCD2*	3.62	3
*GVQW1*	4.05	2	*TMEM218*	0.62	9
*H1FOO*	3.23	3	*TNNC1*	4.08	6
*HMMR*	2.92	8	*TNNC2*	4.08	4
*HNF1A*	1.15	7	*TSPAN12*	2.69	5
*HSPB1*	3.23	3	*WSB1*	2.31	4
*IKZF5*	2.85	5	*ZBED5*	2.46	4
*IL17C*	1.78	4	*ZBED8*	2.83	4
*KIZ*	1.23	6	*ZC3H7B*	1.38	8
*LRRC29*	3.09	4	*ZNF91*	4.88	2

**Table 2 genes-15-00657-t002:** Names and tissue distributions of frequently studied Cadherins.

Familiar Name	Tissue Distribution	HGNC	Type	Has Tunicate Ortholog?
Cadherin E	Epithelial	CDH1	I	Yes
Cadherin N	Neural	CDH2	I	Yes
Cadherin P	Placental	CDH3	I	
Cadherin R	Retinal	CDH4	I	
Cadherin VE	Epithelial	CDH5	II	
Cadherin K	Brain; kidney	CDH6	II	
Cadherin OB	Osteoblast	CDH11	II	Yes
Cadherin BR	Brain	CDH12	II	
Cadherin T and H	Heart	CDH13	I	
Cadherin M	Muscle	CDH15	I	

**Table 3 genes-15-00657-t003:** Latest orthologs of Ephron and Ephrin genes.

Ephron Genes		Ephrin Genes	
HGNC Symbol	Latest Ortholog in:	HGNC Symbol	Latest Ortholog in:
*EPHA2*	Tunicata	*EFNA1*	Tunicata
*EPHA3*	Elasmobranchii	*EFNA2*	Tunicata
*EPHA4*	Tunicata	*EFNA3*	Tunicata
*EPHA5*	Branchiostomata	*EFNA4*	Tunicata
*EPHA6*	Elasmobranchii	*EFNA5*	Tunicata
*EPHA7*	Elasmobranchii	*EFNB1*	Tunicata
*EPHA8*	Elasmobranchii	*EFNB2*	Branchiostomato
*EPHA10*	Osteichthyes	*EFNB3*	Elasmobranchii
*EPHB2*	Branchiostomata		
*EPHB3*	Elasmobranchii		
*EPHB4*	Cnidaria		

**Table 4 genes-15-00657-t004:** Abridgement of DAVID cluster analysis of 36 genes not discussed in main text.

	Term	Genes	FDR *
Cluster 2			
	heterophilic cell-cell adhesion via plasma membrane cell adhesion molecules	*NECTIN3*, *CBLN1*, *NECTIN1*	0.358759
	identical protein binding	*TFAP2E*, *COL23A1*, *HSPB1*, *NECTIN3*, *CBLN1*, *NECTIN1*	0.671885
	Synapse	*NECTIN3*, *CBLN1*, *NECTIN1*	0.935728
Cluster 4			
	homophilic cell adhesion via plasma membrane adhesion molecules	*PALLD*, *NECTIN3*, *NECTIN1*	0.739434
	DOMAIN:Ig-like C2-type 1	*PALLD*, *NECTIN3*, *NECTIN1*	0.656496
	DOMAIN:Ig-like C2-type 2	*PALLD*, *NECTIN3*, *NECTIN1*	0.656496
	Immunoglobulin domain	*PALLD*, *NECTIN3*, *NECTIN1*	0.877976
	Immunoglobulin-like fold	*PALLD*, *NECTIN3*, *NECTIN1*	0.984375
Cluster 6			
	Z disc	*PALLD*, *SYNPO2*, *HSPB1*	0.481232
	actin cytoskeleton	*CALD1*, *PALLD*, *SYNPO2*	0.564483
	actin binding	*CALD1*, *PALLD*, *SYNPO2*	0.614614
	focal adhesion	*PALLD*, *SYNPO2*, *HSPB1*	0.679141
Cluster 7			
	Signaling pathways regulating pluripotency of stem cells	*FZD3*, *ZIC3*, *FZD6*	0.180791
	Developmental protein	*FZD3*, *ZIC3*, *FZD6*, *PITX3*	0.819281
Cluster 8			
	Zinc	*ZMAT1*, *ZIC3*, *ASXL3*	1
	Zinc-finger	*ZMAT1*, *ZIC3*, *ASXL3*	0.988184
	Metal-binding	*ZMAT1*, *ZIC3*, *ASXL3*	1

* False Discovery Rate (maximum is 1).

## Data Availability

No new data were created or analyzed in this study. Data sharing is not applicable to this article.

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
