# Peer review of "Orthologs at the Base of the Olfactores Clade"

_genes, 2024, doi:10.3390/genes15060657_

Round 1
Reviewer 1 Report
Comments and Suggestions for Authors
The manuscript Orthologs at the Base of the Olfactores Clade, authored by Wilfred D. Stein, presents the analysis of the Tunicate orthologous genes found in the human genome. His analysis highlighted that many of the 84 identified genes correspond to human gene families hypothesizing a relevant role of the cadherin proteins in the evolution of the Olfactores clade. Together with other additional contribution related to muscle, nerve and visual perception proteins, the author hypothesizes that ancestor organisms increased the vision capability and the motility, laying the foundations for a more efficient predatory activity of the Vertebrates.
The paper is interesting and full of information. The conclusions are well supported by the data shown, even if the analysis of the evolutionary success of Olfactores based on protein coding genes does not take into account any contributions made by genes whose product is an RNA molecule. This limitation should be added in the conclusions.
Furthermore, the quality of some figures is not great and it would be useful to improve them.
Reviewer 2 Report
Comments and Suggestions for Authors
The manuscript “Orthologs at the Base of the Olfactores Clade” by Wilfred D. Stein evaluates the orthologs of vertebrate genes contained within the tunicate clades. The author has manually curated a dataset of 84 genes most probable to have appeared in the base of the Olfactores clade and describes the biological importance of the evolution of said genes. The manuscript is well written and the results and conclusions solid, but further attention could be given to the figures used.
I have made some comments bellow:
1) From the methods I understood that a complete new search for tunicate orthologs would be done using BLAST, but based on the Results section I understand that this search was done only on the 84 orthologs previously identified. This should be explicit in the Methods section.
2) The methods section does not contain the description on how the phylograms were generated. Also, why are some of the phylograms are rooted at specified genes, but others are not? In the phylograms not rooted in a specific gene, what was the rooting method? Different roots can change the relationship shown in the figures and, therefore, change the interpretation of the results.
3) The author should re-evaluate the need for most figures in the main text. If not essential to the comprehension of the manuscript, the figure should be moved to the Supplementary material or removed altogether. Resolution of the figures should also be improved and some of them seem distorted to fit the page size.
4) As the information of Table 2 and Table 3 is mostly contained within Table S1, I propose the authors remove Tables 2 and 3 and add a column to Table S1 grouping the genes according to their general function (muscle-related, gap-junction, etc).
5) Phylograms could be improved by adding the “ortholog level” next to the gene name, similar to shown in Figure 5 and Figure 23. E.g. “MYL4 (tunicate)” and “MYL2 (corals)”. This would also remove the need for different signs to denote clades (as used in Figures 10 to 12)
6) Either Figure 8 or 9 should be moved to the Supplementary Material or removed altogether, as they are redundant. Also, legend of Figure 9 should be improved to make it independent of Figure 8.
7) In Figure 10, “LEGEND” should be removed.
8) In Figure 12, the phylogram is related to Figure 11 instead of Figure 10. Also, on both phylograms the tip names should be the same.
9) In Figure 14, “A series” should be changed to “EPHA series” and “B series” should be changed to “EPHB series”.
10) A period is missing in line 418, right after “cell-to-cell”.
11) Lines 495-497 are part of Figure 19’s legend.
12) Results section should be renamed as “Results and Discussion” and the “Discussion and Conclusions” should be named “Conclusions”
13) I would like the author had delved deeper in the significance that some of the genes analyzed genes might have had in the evolutionary jump seen in the vertebrates (as commented in the last section for cadherins). This would widen the interest of readers in the paper.
